# Nutraceutical Potential of Djulis (*Chenopodium formosanum*) Hull: Phytochemicals, Antioxidant Activity, and Liver Protection

**DOI:** 10.3390/antiox13060721

**Published:** 2024-06-13

**Authors:** Yu-Chen Huang, Chun-Liang Tung, Shang-Tse Ho, Wei-Sung Li, Shiming Li, Yu-Tang Tung, Jyh-Horng Wu

**Affiliations:** 1Graduate Institute of Biotechnology, National Chung Hsing University, Taichung 402, Taiwan; ychuang8719@dragon.nchu.edu.tw; 2Department of Forestry, National Chung Hsing University, Taichung 402, Taiwan; 3Department of Pathology, Ditmanson Medical Foundation Chia-Yi Christian Hospital, Chiayi 600, Taiwan; 02112@cych.org.tw; 4Department of Wood Based Materials and Design, National Chiayi University, Chiayi 600, Taiwan; stho@mail.ncyu.edu.tw; 5Plant Pathology Division, Taiwan Agricultural Research Institute, Ministry of Agriculture, Taichung 413, Taiwan; weisungli@tari.gov.tw; 6Department of Food Science, Rutgers University, New Brunswick, NJ 08901, USA; shiming@rutgers.edu; 7Advanced Plant and Food Crop Biotechnology Center, National Chung Hsing University, Taichung 402, Taiwan; 8Cell Physiology and Molecular Image Research Center, Wan Fang Hospital, Taipei Medical University, Taipei 116, Taiwan

**Keywords:** *Chenopodium fromosanum* Koidz. (Djulis), Djulis hull, antioxidant activity, phytochemical, lipopolysaccharide, acute liver injury

## Abstract

Djulis (*Chenopodium formosanum*), a traditional Taiwanese crop enriched with phenolic compounds and betalain pigments, is associated with various health benefits, including antioxidant and hepatoprotective effects. This study analysed the phytochemical content and antioxidant capacity of extracts from both the hull and kernel of Djulis. The hull extract, which contained higher levels of flavonoids and exhibited superior antioxidant activity compared to the kernel extract, was selected for further in vivo studies. These experiments showed that oral administration of the Djulis hull crude extract significantly mitigated lipopolysaccharide (LPS)-induced acute liver injury (ALI) in mice by increasing the activity of the antioxidant enzyme glutathione peroxidase (GPx), reducing plasma levels of pro-inflammatory cytokine interferon gamma (IFN-γ), and enhancing liver levels of the anti-inflammatory cytokine interleukin-4 (IL-4). Additionally, the extract demonstrated potential in inhibiting the TLR4/NF-κB pathway, a critical signalling pathway in inflammation and apoptosis, offering insights into its protective mechanisms. These findings underscore Djulis hull’s potential as a functional food ingredient for ALI prevention and propose a valuable application for agricultural by-products.

## 1. Introduction

Acute liver injury (ALI) can be induced by viral infections, medications, excessive alcohol consumption, or sepsis and is a common clinical complication [1]. Severe damage of the liver can lead to acute liver failure (ALF), which is characterized by the rapid onset of coagulopathy and hepatic encephalopathy, along with systemic multiorgan inflammation [2]. In severe cases, ALF can lead to death within days or weeks, with a clinical mortality rate ranging from 25% to 75% [3]. Lipopolysaccharides (LPSs) are among the main components of the outer membrane of Gram-negative bacteria and are used to maintain bacterial structural integrity. Many studies have found that LPSs can induce sepsis and cause multiorgan failure, including liver damage [4,5,6]. The association between LPS-induced endotoxemia and liver injury has been confirmed in animal models. LPS activates liver macrophages, such as Kupffer cells, to release pro-inflammatory cytokines, leading to hepatocyte necrosis or apoptosis and ultimately causing liver damage [7,8,9]. The pathogenesis of sepsis-induced ALI includes inflammation, immune response, cellular hypoxia, apoptosis, and oxidative stress [10].

Kuzmich et al. [11] noted that the TLR4 receptor system detects minute quantities of circulating LPSs (endotoxins) through molecular recognition. Subsequently, receptor dimerization occurs on the cell membrane, triggering a series of protein-protein interactions that generate pro-inflammatory cytokines and interferons. This sequence of events initiates inflammatory and immune responses. Loboda et al. [12] pointed out that when oxidative stress occurs, modifications to the -SH groups of Keap1 and phosphorylation of Nrf2 cause Nrf2 to dissociate from Keap1 and translocate into the cell nucleus; as a result, the transcription of genes such as HO-1 (haem oxygenase 1), GST (glutathione S-transferase), and NQO-1 (NAD(P)H-quinone oxidoreductase 1) is enhanced. Among these, HO-1 can catalyse the production of iron ions (Fe^2+^), biliverdin, and carbon monoxide (CO) from haem. Biliverdin can be converted into the antioxidant bilirubin by biliverdin reductase (BVR), and these products exert protective effects on tissues. Apoptosis is a highly conserved process regulated by the BCL-2 family of proteins, which can promote the death and clearance of damaged and infected cells in response to oxidative stress [13]. Under normal conditions, pro-apoptotic proteins are captured and sequestered by pro-survival proteins of the BCL-2 family. When prosurvival protein binding sites are saturated or missing, BAX and BAK are activated, releasing apoptotic molecules; as a result, caspase-9, caspase-7, caspase-8, and caspase-3 are activated and apoptosis is initiated. Currently, established markers of the apoptosis pathway include BCL-2 family proteins, BAX, cytochrome c, and cleaved caspase [14].

*Chenopodium formosanum* Koidz., also known as Djulis, is a unique species in Taiwan and belongs to the Amaranthaceae family, Chenopodioideae subfamily, and *Chenopodium* genus. Djulis has been cultivated by indigenous people in Taiwan for over 100 years and is a pseudocereal [15]. Djulis possesses various biological effects, such as being an antioxidant, hepatoprotective, skin protector, lowering hyperglycaemia and hyperlipidaemia, and antitumour properties [16,17,18,19], making it suitable for use in health foods. For food processing, Djulis hulls have a bitter taste and poor palatability. Therefore, the hulls are removed before further processing, and a substantial amount of Djulis hull waste is generated annually. Djulis hulls contain a high content of rutin [20], which is known for its hepatoprotective effects [17], ability to prevent hypertension [16], and ability to inhibit fat production [21]. Additionally, compared to Djulis kernels, Djulis hulls contain higher levels of polyphenols and flavonoids, indicating that Djulis hulls have greater potential for biological activity [22]. Previous studies demonstrated the hepatoprotective effects of methanolic extracts from Djulis hulls in a non-alcoholic fatty liver disease mouse model [23], and ethanolic extracts from Djulis hulls have protective effects in LPS-induced RAW 264.7 cell inflammation [22]. This study seeks to delve deeper into the hepatoprotective effects of crude extracts from the Djulis hull and kernel against LPS-induced ALI in mice.

## 2. Materials and Methods

### 2.1. Extraction and Identification of Hulls and Kernels from Three Djulis Strains

Three Djulis strains, red, orange, and yellow, were purchased from Quinoa Green Biotech Co., Ltd. (Taichung, Taiwan). The dehulling process, as well as the separation of hull and kernel, was performed by the same company. The dried Djulis hull and kernel were separately soaked in ethanol at room temperature for 7 d. The extracts were filtered under vacuum, concentrated in a rotary evaporator, and then lyophilized.

The phytocompounds of extracts were analysed using an Orbitrap Fusion Lumos Tribrid mass spectrometer (Thermo Fisher Scientific, San Jose, CA, USA) coupled with a UHPLC (ultra-high-performance liquid chromatography) system (UltiMate 3000 Rapid Separation Dual System, Thermo Fisher Scientific). The method used was based on our previous study [23]. In brief, 10 μL of extract (10 mg/mL in 80% methanol) was subjected to LC-tandem MS analysis. The column oven was maintained at a temperature of 40 °C. The mobile phase, consisting of solvent A (0.1% formic acid in H_2_O) and solvent B (0.1% formic acid in acetonitrile), was delivered at a flow rate of 0.4 mL/min. Separation was achieved using an Acquity UPLC BEH C18 column (1.7 µm, 50 × 2.1 mm; Waters, Milford, MA, USA) with the following gradient: 0–1.0 min, 5% B; 1.0–11.0 min, 5–100% B; 11.0–13.0 min, 100% B; 13.0–13.2 min, 100–5% B; 13.2–15.0 min, 5% B. The mass spectrometry (MS) acquisition settings were as follows: ionization via a heated electrospray ionization (HESI) source with a capillary spray voltage of 3.5 kV, vaporizer temperature set to 275 °C, ion transfer tube temperature maintained at 300 °C, sheath gas at 40 (arbitrary units, A.U.), and auxiliary gas at 8.0 (A.U.). A data-dependent acquisition method was employed for metabolomic analysis, including an initial MS1 spectrum followed by eight MS2 acquisition events targeting the eight most intense ions from the MS1 spectrum in each scan segment. Data were acquired in the positive ion detection mode over the *m*/*z* range of 100–1500. The spectral resolution was set at 30,000 for MS1 and 15,000 for MS2 acquisition, with a minimum ion intensity threshold of 10^6^ (S/N ratio > 5). Higher-energy collision-induced dissociation (HCD, collision energy = 30 eV) was used to facilitate ion activation. For molecular networking analysis, LC-tandem MS data were first converted into the mzXML format using MSconvert (http://proteowizard.sourceforge.net, accessed on 8 March 2022) and subsequently processed with mzXML data through the Global Natural Products Social Molecular Networking website (GNPS, https://gnps.ucsd.edu/ProteoSAFe/static/gnps-splash.jsp, accessed on 8 March 2022). A molecular network was constructed using the online workflow (https://ccms-ucsd.github.io/GNPSDocumentation/, accessed on 8 March 2022) available on the GNPS website. The data processing for molecular networking followed a previously reported protocol [23]. The molecular networks were visualized using Cytoscape 3.9.1. The molecular networks generated from three strains of Djulis hull and kernel crude extracts are publicly accessible at https://gnps.ucsd.edu/ProteoSAFe/status.jsp?task=b3104ce855af480abe2ef1c7e8d12a23 (accessed on 8 March 2022).

### 2.2. Determination of Total Phenolic Contents, Total Flavonoid Contents, and Antioxidant Activity

#### 2.2.1. Total Phenolic Contents

Total phenolic contents were determined according to the Folin–Ciocalteu method, using gallic acid as the standard, followed by Kujala et al. [24] with slight modification. Three strains of Djulis hull and kernel crude extracts were dissolved in methanol/water (50:50, *v*/*v*). The extract solution (100 µL) was mixed with 100 µL of the 1.0 N Folin–Ciocalteu reagent. The mixture was kept for 5 min, which was followed by the addition of 200 µL of 20% Na_2_CO_3_. After 8 min of incubation at room temperature, the mixture was centrifuged for 10 min (12,000 rpm), and the absorbance of the supernatant was measured at 730 nm using an ELISA reader (Tecan, Chapel Hill, NC, USA). The total phenolic contents were expressed as gallic acid equivalents (GAE) in micromoles per gram sample.

#### 2.2.2. Total Flavonoid Contents

Total flavonoid contents were determined by the AlCl_3_ method using rutin as a standard reported by Quettier-Deleu et al. [25] with slight modification. The test samples were dissolved in dimethyl sulfoxide (DMSO). The sample solution (150 μL) was mixed with 150 μL of 2% AlCl_3_. After 10 min of incubation at ambient temperature, the absorbance of the supernatant was measured at 435 nm using an ELISA reader. The total flavonoid contents were expressed as the rutin equivalent (RE) in micromoles per gram sample.

#### 2.2.3. 1,1-Diphenyl-2-picrylhydrazyl (DPPH) Radical Scavenging Activity

DPPH radical scavenging activity was examined according to a method reported by Gyamfi et al. [26] with slight modification. Briefly, 10 μL of the test sample in DMSO was mixed with 90 μL of a 50 mM Tris–HCl buffer (pH 7.4) and 200 μL of a 0.1 mM DPPH–methanol solution. After 30 min of incubation at ambient temperature, the reduction of the DPPH radical was measured by reading the absorbance at 517 nm using an ELISA reader. Ascorbic acid was used as a positive control. The percentage of inhibition was calculated according to the following equation: % inhibition = [(absorbance of control − absorbance of sample)/absorbance of control] × 100.

### 2.3. Animals and Treatments

A total of 26 male specific-pathogen-free (SPF) Bltw:CD1 (ICR) mice, aged six-weeks-old and weighing 28–36 g, were purchased from BioLASCO Taiwan Co., Ltd. (Taipei, Taiwan). The mice were maintained in individually ventilated cages (IVC; BioZone, Ramsgate, UK) under SPF conditions with free access to food and reverse osmosis (RO) water. The environmental conditions were controlled, with a room temperature of 22 ± 2 °C and humidity of 60 ± 5% with a 12-h light/dark cycle. The mice were fed standard laboratory chow (Altromin 1324 pellets, Altromin, Lage, Germany) and provided with aspen bedding (SAFE^®^ select, Rosenberg, Germany). The experimental dosages and the design of the animal model were referenced from previously published studies [23,27,28,29]. The animal experiments were approved by the Institutional Animal Care and Use Committee (IACUC) of National Chung Hsing University (approval No. 111-011^R^).

The mice were randomly assigned to the following groups: a control group, which received an oral gavage of RO water and was injected intraperitoneally (i.p.) with saline on the 28th d; a water group, which received an oral gavage of RO water and was injected i.p. with 15 mg/kg lipopolysaccharide (LPS) (Lipopolysaccharide from *Escherichia coli* O111:B4, Sigma, St. Louis, MO, USA) on the 28th d to induce ALI; a Dh group, which received 300 mg/kg of the red Djulis hull crude extract via an oral gavage for 28 d and was injected i.p. with 15 mg/kg LPS on the 28th d; and a Dk group, which received 300 mg/kg of the red Djulis kernel crude extract via an oral gavage for 28 d and was injected i.p. with 15 mg/kg LPS on the 28th d.

During the experiment (from day 1 to day 28), the mice’s food and water intake, as well as body weight, were measured at intervals of 5 to 8 d to assess whether the samples or experimental procedures had an impact on the mice. On the 29th d, the mice were euthanized using isoflurane 24 h after receiving an intraperitoneal injection of LPS. Blood samples were collected via retro-orbital bleeding using microhaematocrit capillary tubes with heparin sodium. Whole blood (300 μL) was used for white blood cell classification and sent to a certified laboratory (Accuspeedy Medical Laboratory, Tainan, Taiwan) for analysis. The remaining blood was collected in an EDTA tube and centrifuged at 4 °C and 3000 × *g* for 15 min. The plasma was stored at −60 °C for the measurement of pro-inflammatory cytokine levels. Liver perfusion was performed using 1× phosphate-buffered saline (PBS, pH 7.4, Visual Protein Biotechnology, Taipei, Taiwan) to remove blood from the liver. The liver was then blotted with paper towels to remove any excess buffer and then weighed. The largest liver lobe was fixed flat in a cassette and immersed in 10% formalin at room temperature for histological sectioning. A portion of the second largest liver lobe (approximately 0.1 g) was placed in an empty 2.0 mL microcentrifuge tube and reserved for further analyses, including the assessment of antioxidant enzyme activity, peroxidative index, pro-inflammatory and anti-inflammatory cytokine content, and protein expression levels.

### 2.4. Histopathology Analysis

Liver tissue was preserved in 10% formalin, subsequently embedded in paraffin, sliced into 4-μm-thick slices, stained with haematoxylin and eosin, and then imaged at 200× magnification. A clinical pathologist examined the sections using an Olympus BX51 optical microscope equipped with a charge-coupled device camera (Olympus, Tokyo, Japan). The pathologist assessed the Suzuki scores (i.e., the scores of congestion [0: none; 4: severe], vacuolization [0: none, 4: severe], and necrosis [0: none; 4: >60%]), and the average Suzuki scores were calculated to determine the liver injury level.

### 2.5. Measurement of Antioxidant Enzyme Activity and Lipid Peroxidative Index in the Liver

The antioxidant enzyme activities of glutathione peroxidase (GPx) (703102, Cayman Chemical, Ann Arbor, MI, USA) and catalase (CAT) (707002, Cayman Chemical) and the lipid peroxidative index of thiobarbituric acid reactive substances (TBARS) (700870, Cayman Chemical) were measured using the Cayman assay kit. Superoxide dismutase (SOD) (19160-1KT-F, Sigma–Aldrich, St. Louis, MO, USA) activity was measured using a Sigma–Aldrich assay kit.

### 2.6. Enzyme-Linked Immunosorbent Assay (ELISA)

The liver was homogenized in a RIPA buffer containing a 1% proteinase inhibitor. Protein concentrations were measured using a BCA protein assay kit (#23225, Thermo Scientific, Waltham, MA, USA) for the measurement of pro-inflammatory and anti-inflammatory cytokine levels. The quantification of pro-inflammatory and anti-inflammatory cytokine levels in the plasma and liver was conducted using the enzyme-linked immunosorbent assay (ELISA). The pro-inflammatory cytokines analysed included interferon gamma (IFN-γ) (430804, BioLegend, San Diego, CA, USA), interleukin-6 (IL-6) (431304, BioLegend), tumour necrosis factor alpha (TNF-α) (430904, BioLegend), and IL-1β (432604, BioLegend), while the anti-inflammatory cytokines analysed included IL-4 (431104, BioLegend) and IL-10 (431414, BioLegend). The procedures and reagents used in these assays followed the protocols provided by ELISA kits.

### 2.7. Western Blot

The liver was homogenized in a RIPA buffer containing a 1% proteinase inhibitor. Protein concentrations were measured using a BCA protein assay kit (#23225, Thermo Scientific). Forty micrograms of protein was loaded onto 12% SDS-PAGE gel for gel electrophoresis. Subsequently, the proteins were transferred to a PVDF membrane (100 V, 70 min) and blocked (BlockPRO™ Blocking Buffer from Visual Protein Biotechnology) for 1 h. The membranes were then incubated with various primary antibodies (MyD88, IκBα, HO-1, BAX, cleaved caspase-8, cleaved caspase-3, and GAPDH) at 4 °C overnight. After being washed, the membranes were incubated with HRP-labelled mouse (C04001, Croyez Bioscience, Taipei City, Taiwan) or rabbit (#7074, Cell Signaling Technology, Danvers, MA, USA) secondary antibodies for 2 h. In this study, the primary antibodies used were the following: MyD88 antibody (GTX112987, 1:1000, GeneTex, Irvine, CA, USA); IκBα (L35A5) (#4814, 1:1000, Cell Signaling); haem oxygenase 1 antibody (GTX101147, 1:500, GeneTex); anti-BAX antibody (IR93-389, 1:1000, IReal Biotechnology, Hsinchu, Taiwan); anti-caspase-8 cleaved antibody (IR99-409, 1:500, IReal Biotechnology); caspase 3 p17/19 antibody (IR96-401, 1:500, IReal Biotechnology); and GAPDH antibody (GTX100118, 1:5000, GeneTex). Immunoreactive bands were visualized using enhanced chemiluminescence (ECL). The relative protein expression levels were analysed by densitometry using the ImageJ software package (version 1.54f) (Wayne Rasband, Madison, WI, USA). The values were normalized to GAPDH levels in the liver and expressed as fold increases.

### 2.8. Statistical Analysis

GraphPad Prism 9 was used for data visualization and statistical analysis in this study. For the determination of the total phenolic content, total flavonoid content, and DPPH radical scavenging activity, the results are presented here as the mean ± standard deviation (SD) (*n* = 3). Statistical analysis was performed using one-way analysis of variance (ANOVA) followed by a post hoc Tukey’s multiple comparison test. Different letters indicate significant differences between groups (*p* < 0.05). In the animal experiments, the results are presented as the mean ± standard error of the mean (SEM) (*n* = 6). Statistical analysis was performed using a one-tailed Mann-Whitney U test, where ^#^
*p* < 0.05 and ^##^
*p* < 0.01 compared with the control group, and * *p* < 0.05 and ** *p* < 0.01 compared with the water group.

## 3. Results and Discussion

### 3.1. Differences in Composition among the Three Strains of Djulis Hull and Kernel Crude Extracts

The three strains of Djulis hulls and kernels were extracted three times with 95% ethanol. The yields of Djulis hulls (red hull 14.3%, orange hull 12.2%, and yellow hull 12.7%) were all higher than those of Djulis kernels (red kernel 7.3%, orange kernel 7.4%, and yellow kernel 7.8%). Among them, the highest yield was observed in the red hull, followed by the yellow hull and orange hull. Huang et al. (2019) soaked Djulis hull powder in 70% ethanol, shook it (150 rpm) at room temperature for 12 h, and achieved the highest yield (12.4%). In addition, when extraction was performed under high pressure at 600 MPa for 5 min, the yield could reach as high as 18.3%. The extraction yields of Djulis hulls in this study are similar to the yields reported by Huang et al. [22].

As shown in Figure 1, the Cytoscape results for three strains of Djulis hull and kernel crude extracts were numbered 1 to 20 according to the contents of identified components in the database (Appendix A). Most of these components are long-chain, low-polarity compounds, such as triacylglycerols (TAGs) and phosphatidylcholines (PCs). Among the top 20 compounds, **12** (rutin), **18** (kaempferol-3-*O*-rutinoside), and **20** (flavovilloside) belong to the flavonoid group. In addition, the contents of rutin and kaempferol-3-*O*-rutinoside in Djulis hull crude extract were higher than those in Djulis kernel crude extract. Further comparison of flavonoids in Djulis hulls and kernels (Figure 1B and Appendix A) revealed that among the 15 flavonoid compounds, the contents of rutin, kaempferol-3-*O*-rutinoside, quercitrin, NCGC00179918-02, kaempferol-3-*O*-rhamnoside-7-*O*-rhamnoside, kaempferol-3-*O*-pentoxyl-pentoside, and quercetin 3-xylosyl-(1→2)-alpha-l-arabinofuranoside were higher in Djulis hull crude extract than in Djulis kernel crude extract. Therefore, we hypothesized that Djulis hull crude extract shows higher antioxidant potential than that of Djulis kernel crude extract. Additionally, as shown in Figure 1C and Appendix A, the three different strains of Djulis hull crude extracts have similar flavonoid compositions.

Tung et al. [23] found that rutin (quercetin-3-*O*-rutinoside), as a highest-content secondary metabolite in methanolic extracts of Djulis hulls and kernels, as well as quercitrin (quercetin-3-*O*-rhamnoside), hyperoside (quercetin-3-*O*-glucoside), kaempferol-3-*O*-rhamnoside-7-*O*-rhamnoside, and quercetin 3-(2*R*-apiosylrutinoside) (quercetin-3-*O*-apiosyl-(1→2)-rhamnosyl-(1→6)-glucoside), are present in the flavonoid group of Djulis hull crude extract. In addition, previous studies showed that the major compounds of the water extracts of Djulis are betalains and various flavonoids, including rutin and several kaempferol derivatives, such as kaempferol-3-*O*-rutinoside and kaempferol-3,7-di-*O*-rhamnoside (kaempferol-3-*O*-rhamnoside-7-*O*-rhamnoside) [30,31,32]. Hsu et al. [33] identified phenolic acids and flavonoid compounds in 30–70% ethanol to water extracts of Djulis kernels, with flavonoids primarily consisting of quercetin derivatives and rutin. Hence, in Djulis, betalains can solely be acquired via water extraction. In this study, the ethanolic extract did not contain betalains, but other flavonoid compounds were consistent with those previously found in water extract, which were primarily composed of quercetin and kaempferol and their derivatives.

### 3.2. The Total Phenolic and Flavonoid Contents and Antioxidant Activity in Three Strains of Djulis Hull and Kernel Crude Extracts

In this study, the total phenolic and flavonoid contents and antioxidant activities of three strains of Djulis hull and kernel crude extracts were evaluated. As shown in Figure 2A,B, the total phenolic content (red hull, 318.0 ± 2.1 μmole of GAE/g; orange hull, 281.5 ± 9.1 μmole of GAE/g; and yellow hull, 342.9 ± 2.6 μmole of GAE/g) and total flavonoid content (red hull, 32.9 ± 0.4 μmole of RE/g; orange hull, 47.8 ± 0.8 μmole of RE/g; and yellow hull, 40.0 ± 0.4 μmole of RE/g) of Djulis hull crude extracts were significantly higher than those of Djulis kernel crude extracts (total phenolic content: red kernel, 45.4 ± 0.3 μmole of GAE/g; orange kernel, 71.0 ± 2.8 μmole of GAE/g; and yellow kernel, 69.6 ± 4.7 μmole of GAE/g; and total flavonoid content: red kernel, 25.1 ± 0.9 μmole of RE/g; orange kernel, 26.5 ± 0.5 μmole of RE/g; and yellow kernel, 22.6 ± 0.9 μmole of RE/g). In Figure 2D, a lower half-maximal inhibitory concentration (IC_50_ values) for DPPH radical scavenging activity indicated higher antioxidant activity. The results showed that the IC_50_ values of Djulis hull crude extracts (red hull 26.0 ± 0.7 μg/mL, orange hull 38.1 ± 0.9 μg/mL, and yellow hull 30.7 ± 0.3 μg/mL) were significantly lower than those of Djulis kernel crude extracts (red kernel 131.2 ± 4.2 μg/mL, orange kernel 131.9 ± 5.1 μg/mL, and yellow kernel 228.4 ± 4.4 μg/mL). These results indicated that the antioxidant activities of Djulis hull crude extracts were significantly higher than that of Djulis kernel crude extracts, and the red strain of Djulis hull crude extract exhibited the best antioxidant activity (lowest IC_50_ value).

Tung et al. [23] used methanol to extract Djulis hulls and kernels, resulting in a total phenolic content and DPPH radical scavenging activity (IC_50_ values) of 89.6 mg of GAE/g (526.7 μmole of GAE/g) and 32.7 μg/mL for Djulis hulls and 28.7 mg of GAE/g (168.7 μmole of GAE/g) and 324.8 μg/mL for Djulis kernels. As the extraction solvents were different from those used by Tung et al. [23], the total phenolic content was lower in ethanolic extracts, but the DPPH radical scavenging activity of Djulis hull crude extract was similar. Li et al. [15] also used ethanol as the solvent but extracted for 12 h, resulting in a total phenolic content and DPPH radical scavenging activity (IC_50_ values) of 34.6 mg of GAE/g (203.3 μmole of GAE/g) and 291.9 μg/mL for red Djulis hulls and 3.1 mg of GAE/g (18.2 μmole of GAE/g) and 2511.5 μg/mL for yellow Djulis hulls. The shorter extraction time in this study resulted in lower total phenolic content and higher IC_50_ values of DPPH radical scavenging activity. The orange hull contained the highest total flavonoid content, followed by the yellow hull and red hull. Regarding phenolic compounds, the yellow hull had the highest content, followed by the red hull and orange hull. In this study, it was observed that the red and yellow hulls demonstrated better antioxidant capacity in terms of IC_50_ values, namely, red = yellow > orange. It is important to clarify that while flavonoids are a significant group of polyphenols, the antioxidant activity of plant extracts can also depend on the overall composition of polyphenols, including non-flavonoid compounds. Generally, a higher content of phenolic compounds indicates stronger antioxidant capabilities. However, it is also necessary to consider the structural differences between types of phenolic compounds, such as the number of hydroxyl groups (OH) and their positions within the molecule, which are crucial factors affecting their antioxidant effectiveness. In summary, the findings of this study are consistent with those of previous research, demonstrating that Djulis hull crude extracts surpass Djulis kernel extracts in total phenolic content, total flavonoid content, and DPPH antioxidant activity. Among these, the red Djulis hull crude extract exhibits the highest antioxidant activity. Furthermore, the red strain of Djulis, being the predominant strain on the market, was selected for subsequent experiments on ALI in mice.

### 3.3. Effects of Djulis Hull and Kernel Crude Extracts on Food Intake, Water Consumption, Body Weight, and White Blood Cell Classification in LPS-Induced ALI in Mice

In this experiment, an LPS-induced ALI model in male ICR mice was employed to simulate fulminant hepatic failure, which is widely used to investigate preventive treatments for ALI [34]. Previous studies have indicated that LPS can induce oxidative stress and elevate inflammatory responses in the liver, leading to liver function impairment [34,35,36]. Djulis could improve liver damage, including enhancing SOD activity, restoring GSH levels, suppressing lipid peroxidation, and reducing DNA damage in a CCl_4_-induced liver injury model in Wistar rats [17]. Furthermore, a 50% ethanolic extract of Djulis could increase SOD and CAT antioxidant enzyme activities and reduce pro-inflammatory cytokines IL-6 and TNF-α levels in chronic liver injury and fibrosis models in BALB/c mice [20]. The 50% ethanolic extract of Djulis could decrease liver TG and glycerol levels, inhibit lipid peroxidation, and increase GPx and CAT activity in an alcohol-induced fatty liver in C57BL/6J mice [37]. The methanolic extract of Djulis hulls could reduce perirenal white adipose tissue (pWAT), decrease pro-inflammatory cytokine IL-6 and TNF-α protein expression, and enhance GPx and CAT activities in non-alcoholic fatty liver and hyperglycaemia models [23,38]. Based on the composition analysis and antioxidant activity results, Djulis hull crude extracts may exert a preventive effect on LPS-induced ALI. Therefore, we assessed white blood cell classification, liver tissue histological scoring, pro-inflammatory and anti-inflammatory cytokines, antioxidant enzyme activities, and the lipid peroxidation index of Djulis crude extracts on LPS-induced ALI, and the underlying mechanism was determined to explore the TLR4/NF-κB, Nrf2/HO-1, and apoptosis pathways.

Male ICR mice for this experiment were generated over a period of 28 d, and the food intakes, water consumption, and body weights were recorded. After 28 d of feeding, intraperitoneal injection of 15 mg/kg LPS was performed, resulting in significant body weight loss in the LPS-injected groups (water, Dh, and Dk), whereas, the control group, injected with an equivalent volume of saline, maintained the same body weight (Figure 3A). These results revealed that the weight loss observed in the LPS-injected mice solely resulted from the influence of LPS. The changes in body weight for each group are shown in Figure 3B, revealing that all LPS-injected groups experienced significant weight loss along with various symptoms, such as diarrhoea, internal bleeding, and reduced mobility. Twenty-four hours after intraperitoneal injection, only the water group exhibited mouse mortality, while the Dh and Dk groups showed better locomotor activity, less diarrhoea, and less internal bleeding than the water group. As shown in Figure 3C, the liver-to-body weight ratio indicated that all LPS-induced groups (water, Dh, and Dk) showed significantly higher ratios than the control group (4.78 ± 0.08%), with no significant difference among the water, Dh and Dk groups. The increase in liver weight aligns with the trend in the liver-to-body weight ratio observed in a model of ALI induced by LPS/d-GalN reported by Yu et al. [36].

According to the white blood cell classification from Taconic Biosciences, the normal white blood cell concentration in male ICR mice is 6.6625 ± 2.0486 × 10^3^/μL, with neutrophils at 0.954 ± 0.705 × 10^3^/μL (comprising 5.40% to 19.04%), lymphocytes at 5.523 ± 1.710 × 10^3^/μL (comprising 82.64% to 83.03%), monocytes at 0.156 ± 0.142 × 10^3^/μL (comprising 0.3% to 3.42%), eosinophils at 0.030 ± 0.042 × 10^3^/μL (comprising 0% to 0.82%), and basophils at 0 ± 0 × 10^3^/μL (comprising 0%). In the cell frequencies in common samples on the Bio-Rad website, the proportions of cell types in the peripheral blood of mice are as follows: neutrophils (4–6%), T cells (17–20%), CD4^+^ cells (8–12%), CD8^+^ cells (7–10%), B cells (35–58%), NK cells (4–7%), iNKT cells (0.2–0.5%), monocytes (2–3%), and eosinophils (1–2%). Figure 3D displays the white blood cell classification for whole blood in mice, with the control group conforming to the normal white blood cell distribution observed in mice. In contrast, the LPS-induced groups (water, Dh, and Dk) exhibited a significant increase in neutrophil proportions and a significant decrease in lymphocyte proportions compared to the control group. Schwab et al. [39] reported that mice exposed to external stimuli, such as ethanol and corticosterone, exhibited an increase in neutrophil content and a decrease in lymphocyte count. Neutrophils play a role in phagocytosis and bactericidal activity and increase in number during bacterial infections or acute inflammation [40,41,42,43]. Based on the white blood cell classification, ALI was induced in the mice using LPS, which is recognized by the immune system as a Gram-negative bacterial outer membrane component. This induction met the conditions of bacterial infection and acute inflammation, resulting in an increase in neutrophil content and a decrease in lymphocyte content in the blood. Trends towards recovery were observed in the Dh and Dk groups.

O’Connell et al. [44] showed that monocytes are the largest white blood cell type and a primary source of cytokines (such as IL-1β, TNF-α, and IL-6). An increase in monocyte count is associated with bacterial infection, chronic inflammation, and tumour formation. This phenomenon is also reflected in the proportions of monocytes shown in Figure 3D, and significant increases were observed in the Dh and Dk groups compared to the control group. O’Connell et al. [44] also noted that eosinophils and basophils are associated with allergies and parasitic reactions and are relatively less common in mice, which is consistent with the results of this study. The white blood cell classification in this study indicated that the induction of ALI led to an increase in neutrophil and monocyte content and a decrease in lymphocyte content, confirming that ALI was successfully induced. However, it is also evident that Djulis hull and kernel crude extracts had no significant impact on the distribution of white blood cells in response to LPS-induced ALI.

In summary, all mice treated with LPS exhibited a decrease in body weight, an increase in relative liver weight, and significant changes in the distribution of white blood cells. However, no significant improvements were observed in the decline in body weight, increase in relative liver weight, or abnormalities in the distribution of white blood cells with the administration of Djulis hull and kernel crude extracts. This suggests that the dose used in the study was ineffective in alleviating the physiological changes caused by LPS. A possible reason could be that the inflammatory response in the liver caused by LPS treatment was too severe, and the current dose of Djulis hull and kernel crude extracts was insufficient to effectively repair such severe physiological changes. Therefore, future research should consider exploring different doses of Djulis hull and kernel crude extracts to assess whether higher doses can effectively improve the conditions caused by LPS, such as weight loss, increased liver weight, and abnormalities in the distribution of white blood cells.

### 3.4. Djulis Hull and Kernel Crude Extracts Alleviate Liver Histopathological Changes in LPS-Induced ALI in Mice

As shown in Figure 4A, the control group exhibited normal liver tissue, while the water group displayed vacuolization, necrosis, and congestion in liver tissue due to LPS-induced ALI. However, the symptoms were partially alleviated in the Dh and Dk groups. The Suzuki score (Figure 4B–E) was used to assess the vacuolization, necrosis, congestion, and total score in liver tissue sections. The results indicated that the scores for the water group were significantly higher than those of the control group, while no significant differences were observed between the Dh group and the Dk group. However, compared to the water group, the Dh group exhibited significant reductions in vacuolization, necrosis, and total score, while the Dk group showed a significant reduction in necrosis. This result suggested that Djulis hull and kernel crude extracts exhibit a protective effect on liver histopathological changes in ALI induced by LPS.

### 3.5. Evaluation of Djulis Hull and Kernel Crude Extracts on Antioxidant Enzymes and Lipid Peroxidation in the Liver of LPS-Induced ALI in Mice

Based on the liver histopathological changes indicating the significant efficacy of Djulis in attenuating ALI and considering the excellent DPPH radical scavenging activity of Djulis, Djulis may affect the activities of antioxidant enzymes and lipid peroxidation in mouse liver tissue. As shown in Figure 5, the superoxide dismutase (SOD) activity (11.43 ± 1.28 U/mg protein) and glutathione peroxidase (GPx) activity (288.26 ± 19.40 nmol/min/mg protein) in the water group were significantly lower than those in the control group (18.38 ± 3.02 U/mg protein and 390.63 ± 24.74 nmol/min/mg protein). This result indicated that LPS-induced ALI led to a decrease in SOD and GPx activities in the liver, resulting in oxidative stress in the mice. As shown in Figure 5A,B, compared to the water group, the Dh group exhibited a trend of increased SOD activity (19.88 ± 3.63 U/mg protein) and a significant increase in GPx activity (441.75 ± 51.86 nmol/min/mg protein), while a significant increase in GPx activity was observed in the Dk group (418.79 ± 53.03 nmol/min/mg protein). For catalase (CAT) activity, while the water group showed a decreasing trend compared to the control group, no significant differences were observed in LPS-induced groups (Figure 5C). Moreover, based on the DPPH scavenging assay, the antioxidant capacity of the Djulis hull crude extract was also better than that of the Djulis kernel crude extract (Figure 2C,D), indicating that the Djulis hull crude extract performs better in both in vitro and in vivo tests. Additionally, there were no significant differences in thiobarbituric acid reactive substance (TBARS) levels among all groups (Figure 5D). This result suggests that Djulis hull crude extract is more effective than Djulis kernel crude extract in terms of antioxidant enzyme activities. From the summarized results, it is evident that the crude extracts from the Djulis hull and kernel exhibit varied effects on the activities of SOD and GPx. This variation is presumed to be linked to the content of active compounds. Figure 1A presents the Cytoscape analysis of three strains of Djulis, focusing on the hull and kernel crude extracts. It highlights the top 20 compounds by total content, which include flavonoids such as rutin, kaempferol-3-*O*-rutinoside, and flavovilloside. Current research has confirmed that oral administration of rutin at a dose of 100 mg/kg for 45 days significantly decreased fasting plasma glucose levels, increased insulin levels, and enhanced the antioxidant status in diabetic rats [45]. Additionally, mice treated with kaempferol-3-*O*-rutinoside exhibited significantly restored glutathione (GSH) levels and maintained normal CAT and SOD activities in comparison to mice with CCl_4_-induced oxidative liver damage [46]. In the present study, it is notable that the contents of rutin and kaempferol-3-*O*-rutinoside are higher in the hull than in the kernel. This suggests that the Djulis hull, due to its higher contents of rutin and kaempferol-3-*O*-rutinoside, may have enhanced capabilities to regulate CAT and SOD activities.

Chyau et al. [32] noted that pre-treating HepG2 human hepatocellular carcinoma cells with Djulis water crude extract and inducing oxidative stress using *t*-BHP resulted in a significant increase in cellular GSH content, along with a significant decrease in TBARS and reactive oxygen species (ROS) generation compared to cells without pretreatment. Chu et al. [47] showed that the human lung epithelial cell line A549 was pretreated with Djulis water extract and subjected to oxidative stress induced by suspended particles, resulting in a significant increase in SOD activity, increased GSH content, and a notable reduction in TBARS and ROS generation. Chen et al. [31] demonstrated that treating Wistar rats with alcoholic liver injury using Djulis water crude extract resulted in a significant decrease in TBARS content, a significant increase in CAT activity, and no significant differences in SOD and GPx activities. Lin et al. [20] found that administering a 50% ethanolic extract of Djulis to BALB/c mice with CCl_4_-induced chronic liver injury resulted in significantly increased SOD and CAT activities, reduced TBARS content, and no significant change in GPx activity. Based on previous studies, Djulis can mitigate oxidative stress induced by various cell and animal models by enhancing antioxidant enzyme activity, which aligns with the findings of this study.

### 3.6. Evaluation of Djulis Hull and Kernel Crude Extracts on Pro-Inflammatory and Anti-Inflammatory Cytokine Levels in Mouse Plasma and Liver Tissue of LPS-Induced ALI in Mice

Pro-inflammatory and anti-inflammatory cytokines are primarily classified based on the functional polarization of macrophages. Celik et al. [48] noted that macrophage phenotypes can be categorized into the pro-inflammatory M1 phenotype and the anti-inflammatory M2 phenotype, which can undergo polarization (mutual transformation) depending on the surrounding mediators. The M1 phenotype can be induced by pro-inflammatory cytokines and bacterial LPS, and can produce many pro-inflammatory mediators, such as IL-1β, TNF, and NO. In contrast, although the M2 phenotype secretes fewer pro-inflammatory mediators, it produces more anti-inflammatory mediators, such as IL-4 and IL-10. Anti-inflammatory mediators promote the transition of M1 to M2 macrophage phenotypes, thus reducing inflammation, alleviating pain, and contributing to tissue repair. ALI is characterized by liver damage, hepatocyte necrosis, and inflammation induced by chemicals and harmful substances. Researchers generally have believed that pro-inflammatory cytokines secreted by M1 macrophages exacerbate ALI, while M2 macrophages perform functions that promote tissue repair and secrete anti-inflammatory cytokines, aiding in reducing inflammation and alleviating ALI. Therefore, promoting macrophage polarization towards the M2 phenotype and inhibiting the appearance of the M1 phenotype can improve ALI [49].

The results of pro-inflammatory cytokine levels in mouse plasma (Figure 6A–C) show that the levels of IFN-γ, IL-6, and TNF-α in the LPS-induced groups (water, Dh, and Dk groups; water group: 6416.81 ± 2417.86, 27,522.34 ± 7280.42, and 46.62 ± 7.53 pg/mL; Dh group: 1194.40 ± 194.32, 25,586.16 ± 8663.96, and 46.77 ± 6.57 pg/mL; Dk group: 1381.74 ± 484.37, 8593.08 ± 3814.32, and 41.05 ± 7.85 pg/mL) were significantly higher than those of the control group (18.97 ± 18.97, 39.20 ± 15.10, and 1.10 ± 0.58 pg/mL). Compared to the water group, the Dk group showed a significant reduction in IFN-γ and IL-6 levels, while the Dh group only exhibited a significant reduction in IFN-γ levels. These results suggest that LPS-induced ALI results in elevated pro-inflammatory cytokine levels in the plasma. Djulis kernel and hull crude extracts had a significant impact on reducing IFN-γ levels, whereas Djulis kernel crude extracts specifically exhibited a noteworthy effect on IL-6 levels. These differences may be attributed to the Djulis kernel crude extract containing higher contents of flavovilloside, quercetin 3-(2*R*-apiosylrutinoside), hyperoside, isorhamnetin-3-*O*-rutinoside, and xanthorhamnin compared to the Djulis hull crude extract (Appendix A), which contributes to their varied impacts on reducing IL-6. However, the exact reasons require further verification through animal experiments involving the active compounds.

Regarding the measurement of pro-inflammatory and anti-inflammatory cytokine levels in mouse liver tissue (Figure 6D–G), the levels of pro-inflammatory cytokines IFN-γ, IL-1β, IL-6, and TNF-α in the LPS-induced groups (water, Dh, and Dk groups; water group: 197.25 ± 16.50, 596.02 ± 31.94, 266.62 ± 35.33, and 10.37 ± 0.88 pg/mg protein; Dh group: 290.59 ± 44.01, 705.23 ± 90.28, 267.03 ± 25.16, and 18.01 ± 4.27 pg/mg protein; Dk group: 180.17 ± 16.90, 758.32 ± 72.27, 210.47 ± 33.49, and 8.43 ± 0.76 pg/mg protein) were significantly higher than those of the control group (114.69 ± 18.64, 241.62 ± 26.17, 58.52 ± 9.20, and 3.75 ± 0.32 pg/mg protein). There were no significant differences between the Dh and Dk groups compared to the water group. In this study, we found that the Dk group had significantly reduced IL-6 levels in the blood but did not show significant improvement in the liver following LPS-induced damage. This may be the case as the liver is an organ highly active in immune responses, exhibiting particularly strong reactions to LPS. Even if the concentration of IL-6 in the blood is effectively reduced by Djulis kernel crude extract, there may still be high levels of inflammatory activity within the liver, continuing to produce IL-6. For the levels of the anti-inflammatory cytokines IL-4 and IL-10 (Figure 6H,I), the levels in the LPS-induced groups (water, Dh, and Dk groups; water group: 12.82 ± 0.81 and 417.64 ± 24.70 pg/mg protein; Dh group: 19.11 ± 2.91 and 522.30 ± 52.56 pg/mg protein; Dk group: 12.12 ± 1.36 and 430.97 ± 41.52 pg/mg protein) were significantly higher than those in the control group (5.59 ± 0.76 and 188.31 ± 22.75 pg/mg protein), with the Dh group showing significantly higher IL-4 levels than those of the water group. Therefore, when treating the LPS-treated group with Djulis hull crude extract, increases in the levels of IFN-γ, IL-1β, TNF-α, IL-4, and IL-10 in liver were observed, which may be related to the phenotypic transition of macrophages from M1 to M2. M1 macrophages are pro-inflammatory, primarily producing cytokines such as IFN-γ, IL-1β, and TNF-α to participate in the inflammatory response, while M2 macrophages are anti-inflammatory, regulating and mitigating inflammation through the production of cytokines like IL-4 and IL-10. The compounds in Djulis hull crude extract may have the ability to modulate immune responses, not only stimulating M1 macrophages to produce pro-inflammatory cytokines but also promoting the transition to M2 macrophages, thereby increasing the production of IL-4 and IL-10. This phenotypic transition may be facilitated by the modulation of specific signalling pathways. Treatment with Djulis hull crude extract could induce a comprehensive immune response that includes both pro-inflammatory and anti-inflammatory processes, particularly in response to strong immune stimuli like LPS. This transition aids in regulating immune responses, thus effectively controlling pathological conditions. These compounds could be varied and include, but are not limited to, polyphenols and flavonoids, which have anti-inflammatory or immunomodulatory activities. When considering the results of liver histopathological changes in Figure 4, macrophages in the liver were polarized towards the M2 phenotype, suggesting that Djulis hull crude extract exhibit a protective effect on LPS-induced ALI.

### 3.7. Mechanistic Insights into the Prevention Effects of Djulis Hull and Kernel Crude Extracts on ALI in Mice

Considering the known beneficial effects of Djulis hull and kernel crude extracts on liver damage, antioxidant enzymes, and pro- and anti-inflammatory cytokines, we investigated the potential mechanisms underlying their activity in the liver. Figure 7A illustrates the key proteins in the TLR4/NF-κB, Nrf2/HO-1, and apoptosis signalling pathways in the liver. Figure 7B,C depict the TLR4/NF-κB pathway proteins MyD88 and IκBα. Among all groups, there were no significant differences in MyD88 expression. In terms of IκBα expression, the water group showed a decreasing trend compared to the control group, while the Dh and Dk groups exhibited an increasing trend compared to the water group. This suggests that Djulis hull and kernel crude extracts show the potential to reduce IκBα degradation. As shown in Figure 7D, the protein HO-1 in the Nrf2/HO-1 pathway is presented. In the LPS-induced groups (water, Dh, and Dk groups; water: 1.77 ± 0.07, Dh: 1.74 ± 0.15, and Dk: 1.99 ± 0.18), HO-1 levels were significantly elevated compared to those in the control group (1.00 ± 0.08). There were no significant differences between the Dh and Dk groups compared to the water group, but the Dk group showed a slight increasing trend. Figure 7E–G represent proteins associated with apoptosis. Among these, BAX showed no significant differences among the groups. However, compared to the control group, the water group exhibited an increasing trend. In contrast, the Dh and Dk groups showed a decreasing trend compared to the water group. However, the cleaved caspase-8 of the water group (1.70 ± 0.24) showed a significantly higher level than that of the control group (1.00 ± 0.08), while the Dh group exhibited a decreasing trend compared to that of the water group, and there were no significant differences between the Dk group and the other groups. In addition, cleaved caspase-3 showed no significant differences among the groups.

Previous studies have reported similar trends in protein expression in LPS/d-GalN-induced ALI in mice. Jia et al. [50] reported a significant increase in MyD88 and a decrease in IκBα within the TLR4/NF-κB pathway in the induction group, alongside elevated levels of HO-1, cleaved caspase-8, and cleaved caspase-3 in apoptosis-related proteins. Similar trends in BAX protein expression were observed by Peng et al. [51] in a similar LPS/d-GalN-induced mouse model. Lim et al. [29] also demonstrated that BAX and caspase-3 protein levels were significantly higher in an LPS-induced mouse model of ALI. These findings align with the protein expression trends observed in our study.

Regarding the pathways influenced by Djulis, Chyau et al. [32] found that Djulis water crude extract significantly prevented IκBα degradation, increased the Bcl-2/BAX ratio, and reduced intracellular caspase-3 levels in a model of oxidative stress in HepG2 human liver cancer cells. Chu et al. [47] showed that Djulis kernel water extract exerted a significant effect on enhancing the Nrf2/HO-1 pathway in a PM-induced oxidative stress model in A549 cells. In our LPS-induced mouse model of ALI, the protein expression results suggested that Djulis hull crude extract shows the potential to increase IκBα levels in the TLR4/NF-κB pathway, enhance HO-1 in the Nrf2/HO-1 pathway, and reduce cleaved caspase-8 levels in the apoptosis pathway. Based on all the results mentioned above, Figure 8 illustrates that when a LPS induces acute hepatitis in mice, it increases ROS levels in the body, which in turn stimulates the production of pro-apoptotic proteins. This leads to the saturation of binding sites on anti-apoptotic protein BCL-2, activating BAX and causing the release of the apoptotic molecule cytochrome c. Consequently, this sequentially activates caspase-9, caspase-7, and caspase-3 (converting them into cleaved caspase-9, cleaved caspase-7, and cleaved caspase-3), initiating apoptosis. Activated caspase-3/7 not only induces apoptosis but also activates caspase-8 (cleaved caspase-8), which promotes the production of IL-1β. It is hypothesized that the hull of Djulis can enhance the activity of antioxidant enzymes SOD and GPx, potentially reducing ROS levels and possibly inhibiting BAX, thereby reducing the expression and activation of downstream caspase-8, and consequently, decreasing apoptosis. In terms of anti-inflammatory effects, intraperitoneal injection of LPS in mice activates the TLR4/NF-κB pathway due to the binding of TLRs with LPS and pro-inflammatory cytokines, causing the degradation of IκBα, which is bound to NF-κB. This allows NF-κB to enter the nucleus and increase the expression of pro-inflammatory cytokines. The released pro-inflammatory cytokines polarize macrophages towards the M1 phenotype, thus initiating an immune response. It is hypothesized that the hull of Djulis can increase the levels of IκBα, keeping NF-κB in a bound state in the cytoplasm, thereby reducing the levels of pro-inflammatory cytokine IFN-γ in the plasma. Simultaneously, it increases the levels of anti-inflammatory cytokine IL-4 in the liver, promoting the polarization of macrophages in the liver towards the M2 phenotype, which aids in cell repair and protection.

Our study inevitably faces some limitations. Due to considerable individual variability among the mice, the protein expression data from Western blot analyses primarily suggest trends rather than definitive outcomes. To enhance the reliability of these findings, increasing the sample size in future studies would be beneficial. Additionally, we used crude extracts from the Djulis hull and kernel to treat the mice, which complicates the attribution of specific effects to particular compounds within the mixture. Future research should aim to isolate and test the major compounds in these extracts to more definitively establish their effects.

## 4. Conclusions

This study on the valorisation of the Djulis hull for nutraceutical applications provides significant insights into its potential health benefits, particularly in the context of antioxidant activity and liver protection. Djulis, a traditional Taiwanese crop enriched with phenolic compounds and betalain pigments, has been shown to possess a range of biological activities beneficial for health, including antioxidant and hepatoprotective effects. Through comprehensive analysis, this study demonstrates that the Djulis hull extract, which contains higher levels of flavonoids, exhibits superior antioxidant activity compared to the kernel extract. This heightened antioxidant capacity was further corroborated in vivo, where the oral administration of the Djulis hull crude extract significantly mitigated LPS-induced ALI in mice. The protective effects were attributed to the enhancement of antioxidant enzyme activity, reduction in pro-inflammatory cytokine levels, and inhibition of critical inflammatory pathways. The findings underscore the potential of Djulis hull as a functional food ingredient for the prevention of ALI and highlight the value of agricultural by-products in nutraceutical applications. This study not only contributes to the expanding evidence base for the health advantages of traditional crops but also suggests a significant use for Djulis hulls, thereby promoting the sustainable utilization of agricultural resources.

## Figures and Tables

**Figure 1 antioxidants-13-00721-f001:**
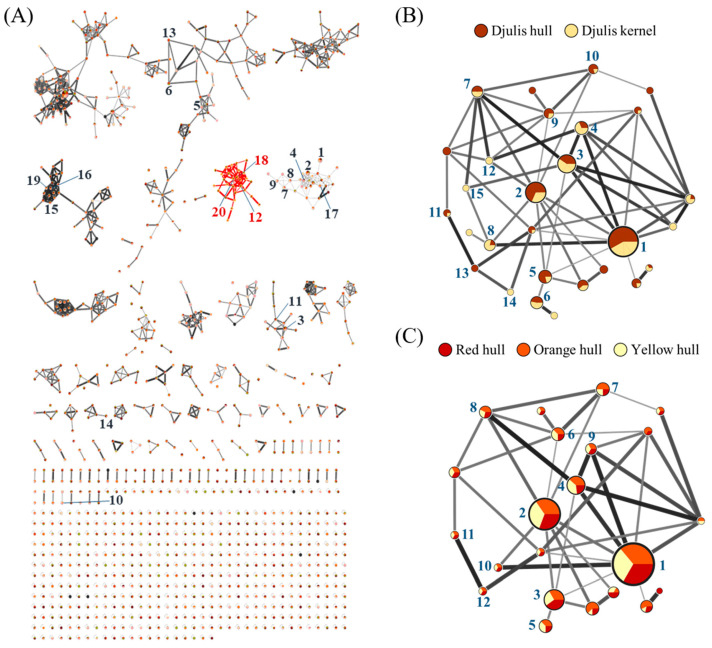
The component analysis of three strains of Djulis hull and kernel crude extracts using UPLC-MS/MS. (**A**) Molecular networking of three strains of Djulis hull and kernel crude extracts. **12**, **18**, and **20** belong to the flavonoid group. (**B**) Differences in flavonoids in Djulis hull and kernel crude extracts. (**C**) Differences in flavonoids in three strains of Djulis hull crude extracts. The widths and shades of connecting lines represent the similarity between connected nodes (Edge Score), while the sizes of circular nodes represent the total spectra number for each compound (spectra number). The numbers in the figures corresponded to the compound name, parent mass, and spectra number, as detailed in Appendix A.

**Figure 2 antioxidants-13-00721-f002:**
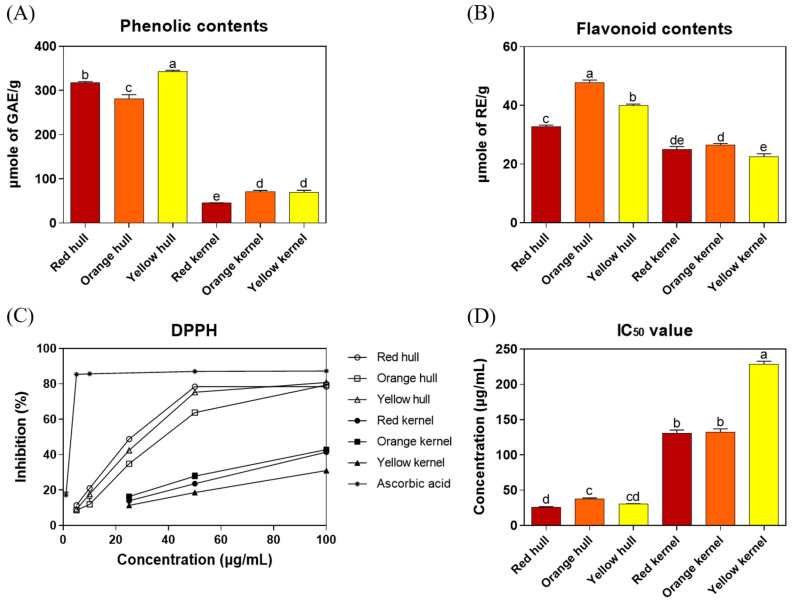
(**A**) Total phenolic contents, (**B**) total flavonoid contents, (**C**) DPPH radical scavenging activity, and (**D**) half maximal inhibitory concentrations of three strains of Djulis hull and kernel crude extracts. DPPH, 1,1-diphenyl-2-picrylhydrazyl. The statistics were determined by one-way ANOVA with Tukey’s multiple comparisons test. Values are represented as the mean ± SD (*n* = 3), and different letters indicate significant differences (*p* < 0.05).

**Figure 3 antioxidants-13-00721-f003:**
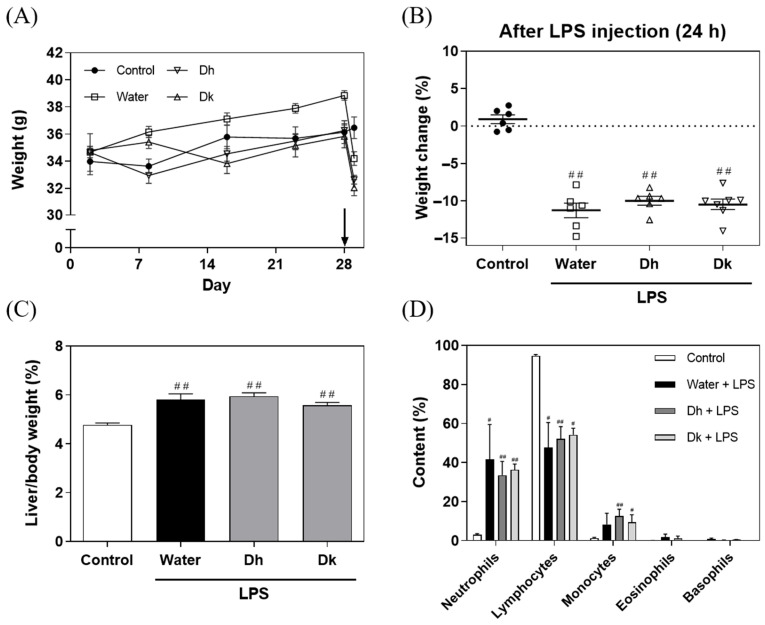
Effects of Djulis hull and kernel crude extracts on (**A**) body weight, (**B**) body weight change, (**C**) relative liver weight, and (**D**) white blood cell type of mice in LPS-induced acute liver injury. The arrow represents the time point induced by LPS. A one-tailed Mann-Whitney U test was used for statistical analysis. Values are represented as the mean ± SEM (*n* = 6), where ^#^
*p* < 0.05 and ^##^
*p* < 0.01 compared with the control group.

**Figure 4 antioxidants-13-00721-f004:**
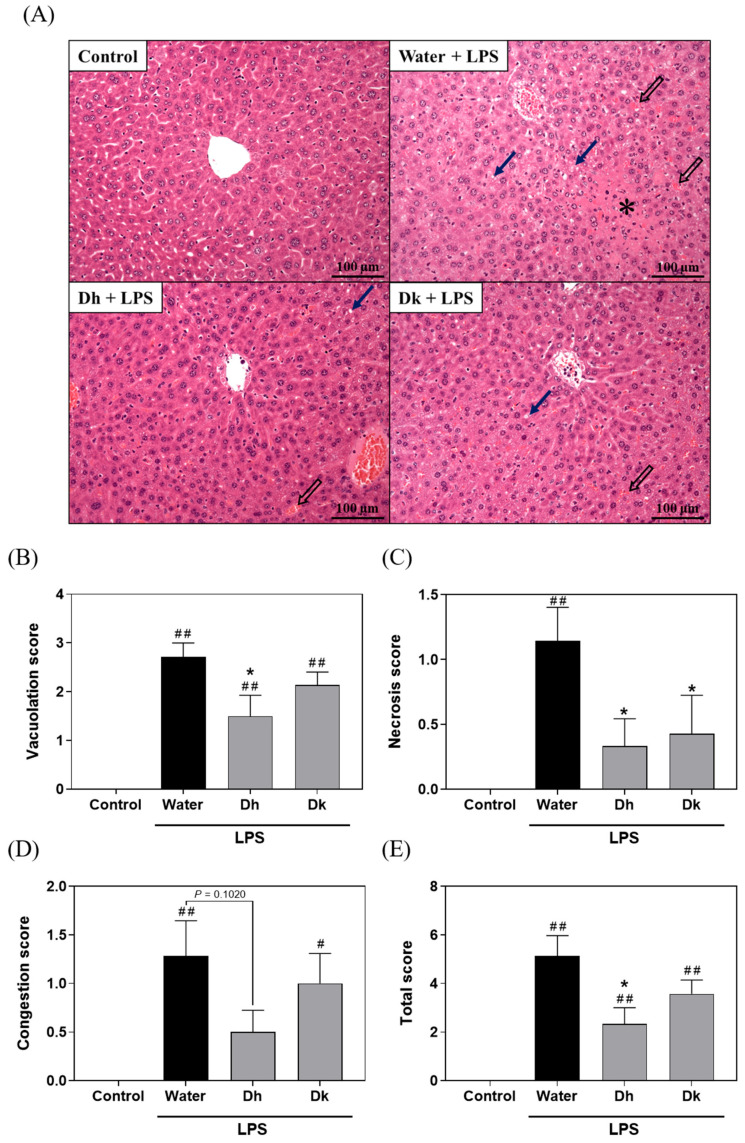
Effects of Djulis hull and kernel crude extracts on the (**A**) liver histopathology and Suzuki scores of mice with an LPS-induced acute liver injury, including (**B**) vacuolation, (**C**) necrosis, (**D**) congestion, and (**E**) the total score. H&E staining (200×). Solid arrows represent vacuolization, * represents necrosis, and hollow arrows represent congestion. A one-tailed Mann-Whitney U test was used for statistical analysis. Values are represented as the mean ± SEM (*n* = 6), where ^#^
*p* < 0.05 and ^##^
*p* < 0.01 compared with the control group, and * *p* < 0.05 compared with the water group.

**Figure 5 antioxidants-13-00721-f005:**
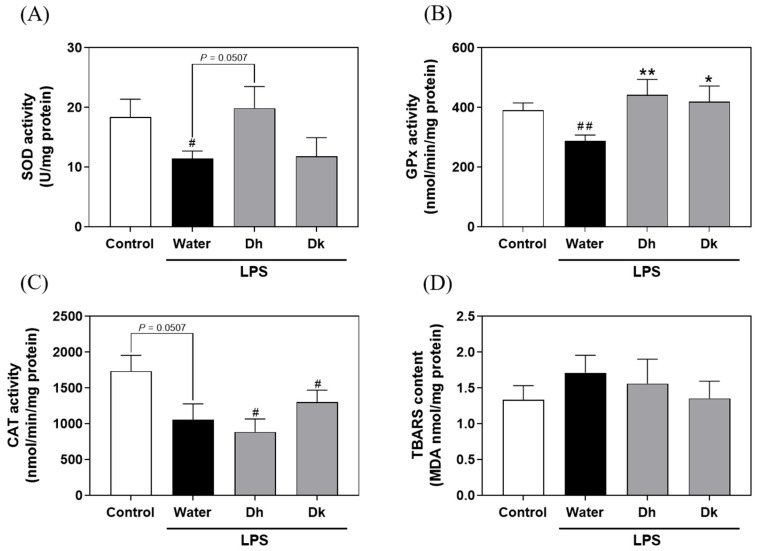
Effects of Djulis hull and kernel crude extracts on the activities of liver antioxidant enzymes, including (**A**) SOD, (**B**) GPx, and (**C**) CAT, as well as the (**D**) TBARS content of mice in LPS-induced acute liver injury. SOD, superoxide dismutase; GPx, glutathione peroxidase; CAT, catalase; TBARS, thiobarbituric acid reactive substances. A one-tailed Mann-Whitney U test was used for statistical analysis. Values are represented as the mean ± SEM (*n* = 6), where ^#^
*p* < 0.05 and ^##^
*p* < 0.01 compared with the control group, and * *p* < 0.05 and ** *p* < 0.01 compared with the water group.

**Figure 6 antioxidants-13-00721-f006:**
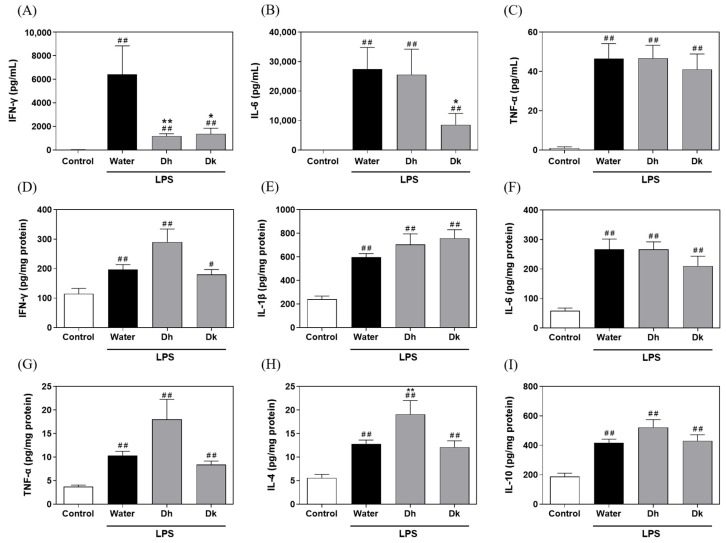
Effects of Djulis hull and kernel crude extracts on plasma pro-inflammatory cytokines (**A**) IFN-γ, (**B**) IL-6, and (**C**) TNF-α and liver pro-inflammatory cytokines (**D**) IFN-γ, (**E**) IL-1β, (**F**) IL-6, and (**G**) TNF-α and liver anti-inflammatory cytokines (**H**) IL-4 and (**I**) IL-10 in mice with LPS-induced acute liver injury. IFN-γ, interferon gamma; IL-6, interleukin-6; TNF-α, tumour necrosis factor alpha; IL-1β, interleukin-1 beta; IL-4, interleukin-4; IL-10, interleukin-10. A one-tailed Mann-Whitney U test was used for statistical analysis. Values are represented as the mean ± SEM (*n* = 6), where ^#^
*p* < 0.05 and ^##^
*p* < 0.01 compared with the control group, and * *p* < 0.05 and ** *p* < 0.01 compared with the water group.

**Figure 7 antioxidants-13-00721-f007:**
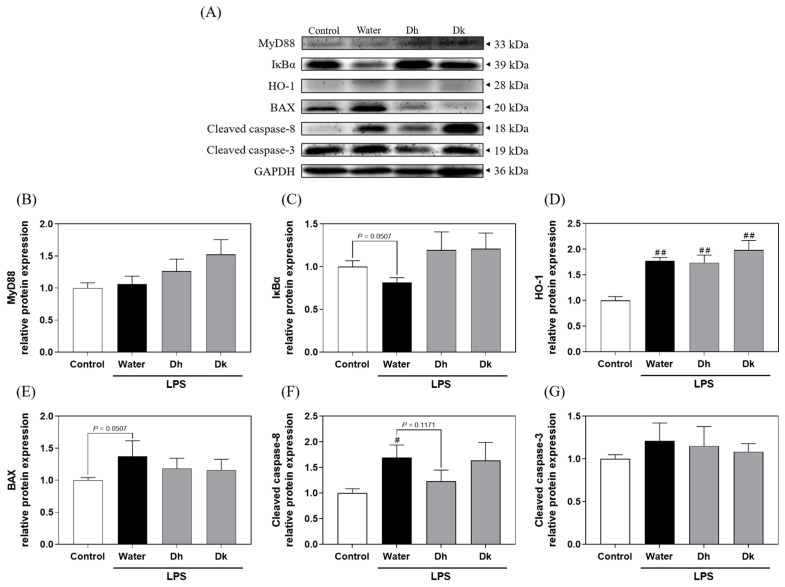
(**A**) Effects of Djulis hull and kernel crude extracts on the protein expression levels of (**B**) MyD88 and (**C**) IκBα in the TLR4/NF-κB pathway, (**D**) HO-1 in the Nrf2/HO-1 pathway, and (**E**) BAX, (**F**) cleaved caspase-8, and (**G**) cleaved caspase-3 in the apoptotic pathway in the livers of mice with LPS-induced acute liver injury. MyD88, myeloid differentiation primary response protein 88; IκBα, nuclear factor-kappa B inhibitor alpha; HO-1, haem oxygenase-1; BAX, Bcl2-associated X protein. A one-tailed Mann-Whitney U test was used for statistical analysis. Values are represented as the mean ± SEM (*n* = 6), where ^#^
*p* < 0.05 and ^##^
*p* < 0.01 compared with the control group.

**Figure 8 antioxidants-13-00721-f008:**
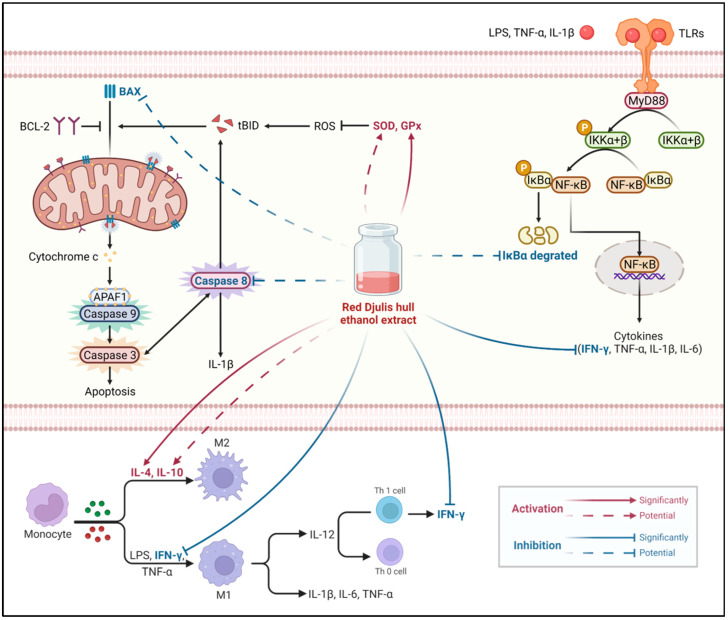
Mechanisms underlying the effects of Djulis hull crude extract on the LPS-induced acute liver injury model in mice (created with BioRender.com). Red solid arrows indicate a significant increase; red dotted arrows indicate a potential increase; blue solid arrows indicate a significant decrease; blue dotted arrows indicate a potential decrease [11,12,13,14,49,52,53].

## Data Availability

The data in this study are available from the corresponding author upon reasonable request.

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
