# Peer review of "Nutraceutical Potential of Djulis (Chenopodium formosanum) Hull: Phytochemicals, Antioxidant Activity, and Liver Protection"

_antioxidants, 2024, doi:10.3390/antiox13060721_

Round 1

Reviewer 1 Report

The topic of the paper is really interesting and athe research is design and conducted very well. However I consider is is necessary to explain certain aspects inorder to get publish the paper:

1-The title needs to be reduced

2-The introduction needs to be extended

3-Methods. Important to asnwer to the following questions:

1-Why the antioxidant capacity was evaluated using DPPH instead of other parameter such as ABTS or FRAP?

2-Give more explanationr egading the antiinfalmatory method is not clear the sampling and method used to evaluation.

3-The results and discussion is really good and well prepared, the graphical abstract close to the conclusion is clear and explain the issues reached clearly.

Really I consider the paper is very good is appropiate for the journal but different aspect mentioned at the top of the section needs to be consider to close loop.

Figures and tables are well preapred and describe the results obtained.

Congratulations for the paper.

The topic of the paper is really interesting and athe research is design and conducted very well. However I consider is is necessary to explain certain aspects inorder to get publish the paper:

1-The title needs to be reduced

2-The introduction needs to be extended

3-Methods. Important to asnwer to the following questions:

1-Why the antioxidant capacity was evaluated using DPPH instead of other parameter such as ABTS or FRAP?

2-Give more explanationr egading the antiinfalmatory method is not clear the sampling and method used to evaluation.

3-The results and discussion is really good and well prepared, the graphical abstract close to the conclusion is clear and explain the issues reached clearly.

Really I consider the paper is very good is appropiate for the journal but different aspect mentioned at the top of the section needs to be consider to close loop.

Figures and tables are well preapred and describe the results obtained.

Congratulations for the paper.

Reviewer 2 Report

The manuscript entitled “Valorization of Djulis (Chenopodium formosanum) Hull for Nutraceutical Applications: Phytochemical Characterization, Antioxidant Activity, and Protective Effects Against Acute Liver Injury” by Yu-Chen Huang et al examined various biological activities of hull and kernel of different Djulis. The authors indicated the importance for the usage of hull, which is the industrial waste, as biological sources.

The authors carried out many experiments to explain the priority of hull than kernel.

The trial is half-done, however, there are several drawbacks found in the manuscript.

1, The authors should be very careful to examine the manuscript to follow the journal format. For example, in the title, the authors used against as a capital letter, while of and for as normal letter.

2, In the reference part, the font size is bigger than in the manuscript.  Is it recommended to do so?

3, There are some grammatical mistakes in the manuscript, for example, line no. 240, extracted is revised to extraction. It is careful to pay much attention not to make primitive mistakes.

4, The authors quoted many references which some authors included in the references, which will misunderstand for readers to self-citation. For example, in the experimental part, the authors cited their literatures in the total phenolic contents (ref. 20), and also total flavonoid and DPPH radical scavenging experiments (reference 21). The original references are not shown in the experimental section. I guess, the authors examined the experiments based on the previous literature (reference should be cited) with slight modification. The authors should be careful for this point (the original work is not belonging to authors).

5, In the Results and Discussion part, the authors quoted many literatures to strengthen their logic, however, the precise discussion was not performed based on their original experimental results. For this reason, it is difficult to accept the conclusion (Fig. 7).

Figures

In Fig. 1, the authors measured polyphenol and flavonoids contents and DPPH radical scavenging activities of red, orange and yellow hulls and red, orange, and yellow kernels. The hull contents of polyphenols and flavonoid contents are higher than those of kernel in every case. Comparing red, orange and yellow hulls, polyphenol contents is highest in yellow following red and orange, while flavonoid content is highest in orange following yellow and red. In the DPPH radical scavenging activity, hulls is stronger scavenging activities than those of kernels. The lowest IC50 value is observed in the case of red hull, which is not consistent with the polyphenol content and flavonoid content. Some explanation is necessary.

In Fig. 2, the authors examined the body change, and white blood cell classification in the presence or absence of red hull and kernel. The body weight was decrease in all LPS-treated mice, however, significant changes (increase or decrease) of liver volume and white blood cell type of mice were not observed in LPS-treated samples. Are the results coming from the experimental designs? Based on these results, the hull and kernel did not provide positive effect for mice. Some explanation is necessary.

In Fig. 3, the authors analyzed histochemical results of four samples (control, water-LPS, Dh-LPS, and Dk-LPS). By the addition of Dh and Dk, the score became better than those of water-LPS sample, especially in the case of necrosis score. The significant difference was observed between Dh-LPS and water. Is a significant difference observed in Dh-LPS and Dk-LPS? The results showed the similar trend of Dh -LPS and Dk-LPS, the Dh-LPS is a little better than Dk-LPS in every case. So, is the difference within the measurement error?

In Fig. 4, the authors measured the antioxidant enzyme activity and TBARS content by LPS treated Mice. By LPS treatment (water) the antioxidant activity of SOD, GPx, and CAT decreased compared to the control group. In the case of Dh, SOD and GPx showed the increased trend, while CAT showed the decreased trend to the control sample. In the case of Dk, SOD and CAT activities showed the decreased trend, while GPx showed the increase trend. There is a clear difference between Dh and Dk administrated mice. Some explanation is necessary. In the case of TBARS, there is no clear difference between all samples examined. The authors described the difference of DPPH radical scavenging activity of Dh and Dk (in Fig. 1 C, D). The results obtained here is not in the same line. Some explanation is necessary.

In Fig. 5, the authors measured various cytokine levels in blood and liver samples of LPS treated mice.  In blood samples (Fig, 5A-C), LPS-treated showed higher levels of IFN-gamma, IL-6, and TNF-alpha, a significant difference was observed in the case of IL-6. The administration of Dk showed lower level than those of Dh and water samples. No clear explanation for this is not shown in the manuscript. In the liver case, the Dh sample showed the increase level of IFN-gamma and TNF-alpha. In both cases, the values were higher than those of water sample. It is necessary to explain for these results. In the case of IL-6, the difference of Dk sample and other two samples is not clear (almost same level), this is different from the blood sample. Some explanation for this is required. In the cases of IL-4 and IL-10, significant difference (increased level) was observed in Dh sample toward water and Dk samples. LPS-treated all samples showed the higher (significant difference) than that of the control sample in IL-4 and IL-10 cases. Based on these experimental results, precise explanation of Dh sample must be necessary, namely, it induces higher IFN-gamma, IL-1beta, IL-6, and TNF-alpha, while it induces higher IL-4 and IL-10.

In Fig. 6, the authors carried out protein expression experiments, however, the results were not clear. The difference of water sample and Dh, Dk samples were not observed (no significant different results). It is difficult to persuade readers based on these results.

The supplementary Fig. 1 is interesting enough to be listed on the manuscript. It will become clearer for readers to know the difference of different hulls and kernel.

Tables

Three tables are summarized into 1 table and included in the manuscript for the better understanding of readers. The chemical structures of characteristic compounds should be shown.

The manuscript is potentially interesting topic, “waste becomes resource”. However, there are many drawbacks. The authors should discuss deeply their experimental results precisely by cross-talking by using the Figure and Figure networking. The logic lacks the strength to persuade readers to agree.

Round 2

Reviewer 1 Report

All the comments mentioned have been addressed.

All the comments mentioned have been addressed.

Author Response

Thank you for your positive feedback on our manuscript.

Reviewer 2 Report

In the revised manuscript entitled “Nutraceutical Potential of Djulis (Chenopodium formosanum) Hull: Phytochemicals, Antioxidant Activity, and Liver Protection” , Yu-Chen Huang et al revised the manuscript according to the suggestions and comments of reviewers.

Figures

In Fig. 1, the reviewer pointed out the flavonoid content is highest in orange hull (significant difference to other ones, red hull is lower than orange, and yellow hulls), while, polyphenol content is highest in yellow hull (significant difference to other ones, orange hull is lower than red, and yellow hulls). In Fog. 1D, red hull showed the lowest IC50 value (almost same with yellow). According to the author’s reply, “total phenolic compounds is usually higher than that with flavonoids alone.--- These results clearly demonstrate a positive correlation between phenolic content of hulls of different colors and their antioxidant capacity” is described.  The reviewer is wondering that the authors pointed out the importance of polyphenols in the antioxidant activity (DPPH radical scavenging), if so, the content of flavonoids does not work on the antioxidant activity. Further, the polyphenol content of yellow hull has a significant difference to red hull, and it is also the case of flavonoid content (in both cases, yellow hull > red hull). A clear explanation for this might be necessary.

In Fig. 2, the authors described “This suggests that the dose used in the study was ineffective in alleviating the physiological changes by LPS. A possible reason could be that the inflammatory response in the liver caused by LPS treatment was too sever, and the current dose of Djulis hull and kernel crude extracts was insufficient to effectively repair such severe physiological changes.” The authors recognized the importance to reexamine the experiments. It is highly desirable to do so.

In Fig. 4, the authors answered as follows “---Moreover, based on the DPPH scavenging assay, the antioxidant capacity of Djulis hull crude extract was also better than that of Djulis kernel crude extract, indicating hull crude extract performs better in both in vitro and in vivo tests. These differences may be due to different proportions of active components in Djulis hull and kernel crude extracts, which differently regulate activities of SOD and GPx”. The authors recognized the importance of active components. So, what is the active components, which the authors think. Are they polyphenol derivatives or not?

In Fig. 5, the authors also answered as follows”—These differences may arise from varying proportions of active components in the crude extracts of Djulis hull and kernel, leading to their differing effects on reducing IL-6.” In addition, the authors answered as follows”---The compounds in Djulis hull crude extract may have the ability to modulate immune responses, not only stimulating M1 macrophages to produce pro-inflammatory cytokines but also promoting the transition to M2 macrophages, thereby increasing the production of IL-4 and IL-10.” Are the active components and compounds in Djulis to modulate immune responses same or not? What type of compounds do the authors imagine?

In Fig. 6, the authors answered “—Significant individual viability among mice means that the protein expression results from Western blot analyses can only indicate trends. Therefore, increasing the number of animal samples in future studies may produce more convincing results.” However, the use of another method might provide good information (qPCR or other methods). If the authors will carry out future works, the authors will try to challenge more quantitative method to evaluate the protein expression quantitatively.

As written in the previous review, the manuscript is potentially interesting topic, “waste becomes resource”. However, the trial is quite primitive and premature. I want something eye-catching in the manuscript.  

Round 3

Reviewer 2 Report

In the second revised version of the manuscript entitled “Nutraceutical Potential of Djulis (Chenopodium formosanum) Hull: Phytochemicals, Antioxidant Activity, and Liver Protection” , Yu-Chen Huang et al revised the manuscript according to the suggestions and comments of reviewers.  The reviewer indicated several points to revise and authors revised the manuscript as much as possible. Although the manuscript has still some weak points, which the authors recognize and write in the letter.

Further works based on the improvement of experimental techniques and discussion are expected in the next manuscript.

As wrote in the general comments, usage of more advanced techniques (qPCR, metabolome analyses, spectroscopic analyses) is highly desirable in future manuscript.

Some editorial English and format check are necessary before acceptance of the manuscript. (editor decision)